# Prevalence of diabetic retinopathy and vision-threatening diabetic retinopathy in adults with diabetes in China

Xuhong Hou[1,14], Limin Wang[2,14], Dalong Zhu[3], Lixin Guo[4], Jianping Weng [5], Mei Zhang[2], Zhiguang Zhou[6], Dajin Zou[7], Qiuhe Ji[8], Xiaohui Guo[9], Qiang Wu [10], Siyu Chen[11], Rong Yu[1], Hongli Chen[1], Zhengjing Huang[2], Xiao Zhang[2], Jiarui Wu[12,13], Jing Wu[2] ✉, Weiping Jia [1] ✉ & for the China National Diabetic Chronic Complications (DiaChronic) Study Group*

The current epidemic status of diabetic retinopathy in China is unclear. A national prevalence survey of diabetic complications was conducted. 50,564 participants with gradable non-mydriatic fundus photographs were enrolled. The prevalence rates (95% confidence intervals) of diabetic retinopathy and vision-threatening diabetic retinopathy were 16.3% (15.3%–17.2%) and 3.2% (2.9%–3.5%), significantly higher in the northern than in the southern regions. The differences in prevalence between those who had not attained a given metabolic goal and those who had were more pronounced for Hemoglobin A1c than for blood pressure and low-density lipoprotein cholesterol. The participants with vision-threatening diabetic retinopathy had significantly higher proportions of visual impairment and blindness than those with non-vision-threatening diabetic retinopathy. The likelihoods of diabetic retinopathy and vision-threatening diabetic retinopathy were also associated with education levels, household income, and multiple dietary intakes. Here, we show multi-level factors associated with the presence and the severity of diabetic retinopathy.

Diabetic retinopathy (DR), a major specific diabetic microvascular complication, occurs in approximately one-third of patients with diabetes[1]. Although DR is often insidious and asymptomatic at the early stages, it might quickly progress into vision-threatening DR (VTDR) without awareness and intervention, and then could lead to irreversible vision impairment. About one-third of patients with proliferative DR (PDR) combined with high-risk characteristics will progress into severe vision loss within three years if not treated promptly[2]. In many countries, DR is a leading cause of preventable vision impairment and blindness in the working-age population[3]. Meanwhile, DR was the only one of the top five causes of blindness that had a globally increased age-standardized prevalence in adults aged 50 years and older between 1990 and 2020, and its prevalence was projected to

continue rising, with an increasing number of patients with diabetes and longer life expectancy[4]. Vision impairment and blindness severely affect the patient's quality of life, increase the incidence risks of comorbidities[5], and reduce life expectancy. However, early systematic screening and timely treatment for DR have been shown to be highly effective in avoiding vision impairment and blindness[6]. With the largest number of people with diabetes, around one-fourth of the global number, living in China, there is a lack of current data representing a national distribution of DR to guide the prevention and control strategy[7].

As we know, so far only two multiple province-level (12-province and 6-province) prevalence surveys of DR were conducted in China[8,9]. However, the two studies lacked a sampling design and their

A full list of affiliations appears at the end of the paper. *A list of authors and their affiliations appears at the end of the paper. ✉e-mail: wujing@chinacdc.cn; wpjia@sjtu.edu.cn

participants were recruited from both hospitals and communities, which resulted in the samples being unrepresentative of Chinese people with diabetes. In recent years, significant changes in factors related to DR, including socio-economic development, lifestyle, dietary patterns, retinal imaging, and treatment, may have affected the epidemiologic features of DR[10,11]. For these reasons, the experts call for continuous high-quality population-based studies[12], and updated nationally representative surveys are also urgently needed to guide the prevention of vision impairment and blindness among patients with diabetes in China.

In this work, we report the distributions of any DR and VTDR nationwide as well as potentially associated factors regarding demographics, geographical regions, socio-economic status, lifestyle factors, and clinical characteristics among Chinese adults with diagnosed diabetes, based on the national survey of diabetic complications in China between 2018 and 2020. An in-depth understanding of the related factors of DR help promote better medical care, a healthier lifestyle, and potential causal research.

## Results

### Characteristics of study participants

For the 50,564 participants, the median (25th percentile-75th percentile) age and diabetes duration were 57.5 years (50.9–64.8) and 5.2 years (2.5–10.1), respectively. Of these participants, 50.3% were females, 49.2% resided in the northern regions, and 46.8% resided in rural areas. The treatment rates for hyperglycemia, hypertension, and dyslipidemia were 78.6%, 39.1%, and 12.3%, respectively, and the corresponding attainment rates of Hemoglobin A1c (HbA1c), Blood pressure (BP), and low-density lipoprotein cholesterol (LDL-C) targets were 44.0%, 29.8%, and 34.9%, respectively (Table 1). Compared with the participants without DR, those with any DR had significantly higher proportions of northerners and longer diabetes duration, but lower education and income levels. The participants with any DR had a higher treatment rate for lowering glucose and comparable treatment rates for hypertension and dyslipidemia, but they still had higher levels of blood glucose, systolic blood pressure, and LDL-C (all $p < 0.050$) (Table 1). Furthermore, similar linear trends were also observed for the above-mentioned characteristics as the severity of DR increased in Supplementary Table 1. Notably, only 13.6% of the 8559 participants with any DR (9.4% of participants with non-VTDR and 32.8% of participants with VTDR) reported a history of DR (Table 1 and Supplementary Table 1).

In addition, there were differences in lifestyle factors, including physical activity, unhealthy behaviors (smoking and alcohol drinking), and multiple diet intakes between the two comparison groups (Table 1).

### Prevalence of DR and VTDR

The weighted prevalence of any DR and VTDR among Chinese adults with diabetes grouped by demographic factors and diabetes duration, as well as by the attainment of metabolic targets, were separately shown in Table 2 and Table 3.

In the patients with diabetes aged 18–74 years, the overall prevalence of any DR was 16.3% (95% CI 15.3%–17.2%), including the prevalence of 0.75% (95% CI 0.64%–0.86%) for diabetic macular edema (DME); and the prevalence of VTDR was 3.2% (95% CI 2.9%–3.5%), including 2.3% (95% CI 2.1%–2.6%), 0.64% (95% CI 0.55%–0.73%), and 0.54% (95% CI 0.45%–0.63%) for severe non-proliferative DR (NPDR), PDR and clinically significant macular edema (CSME), nationwide, respectively. It was estimated that roughly 19.5 million and 3.8 million adults with diagnosed diabetes had any DR and VTDR, respectively, in China (Table 2).

Among the adults with diabetes, the inter-subgroup differences in the prevalence of any DR and VTDR between men and women or between urban residents and rural residents did not reach statistical

significance. The prevalence of any DR and VTDR was significantly higher in the northern than in the southern regions (DR: 18.1% [95% CI 16.6%–19.6%] vs. 14.4% [95% CI 13.3%–15.5%]; VTDR: 3.8% [95% CI 3.4%–4.3%] vs. 2.5% [95% CI 2.2%–2.8%]). The prevalence of any DR and VTDR was 8.2% (95% CI 6.7%–9.7%) and 1.24% (95% CI 0.49%–2.00%) among those with diabetes duration of less than one year and climbed to 38.6% (95% CI 35.8%–41.4%) and 13.8% (95% CI 11.9%–15.6%) among those with diabetes duration of longer than 20 years, respectively (Table 2).

The differences in proportions of any DR and VTDR were statistically significantly higher in those with worse metabolic control versus those with better control (for HbA1c, any DR: 22.3% [95% CI 21.1%–23.4%] vs. 8.7% [95% CI 7.8%–9.6%], VTDR: 4.6% [95% CI 4.2%–5.0%] vs. 1.4% [95% CI 1.1%–1.6%]; for BP, any DR: 17.5% [95% CI 16.5%–18.5%] vs. 13.7% [95% CI 12.5%–14.9%], VTDR: 3.4% [95% CI 3.1%–3.7%] vs. 2.7% (95% CI 2.3%–3.1%); for LDL-C, any DR: 16.7% [95% CI 15.7%–17.8%] vs. 15.4% [95% CI 14.3%–16.5%]) (Table 3).

For the 31 provinces in mainland China, the standardized province-specific prevalence of any DR and VTDR among the adults with diabetes ranged from 9.9% (95% CI 8.3%–11.5%) for Guizhou to 29.1% (95% CI 26.5%–31.6%) for Shandong, and 1.27% (95% CI 0.61%–1.94%) for Jiangxi to 6.3% (95% CI 5.1%–7.5%) for Heilongjiang, respectively. The top 3 provinces for the prevalence of any DR or VTDR, all in the northern regions, were Shandong, followed by Heilongjiang (27.0% [95% CI 24.5%–29.5%]) and Henan (24.6% [95% CI 22.3%–26.8%]) for any DR, and Heilongjiang, Shandong (6.2% [95% CI 5.0%–7.3%]), and Beijing (4.7% [95% CI 3.6%–5.9%]) for VTDR. Of note were the obvious inconsistencies between province-specific prevalence ranks of any DR and VTDR for individual provinces, such as Qinghai (Rank 4 of DR vs. Rank 23 of VTDR) (Table 4).

### Visual impairment and blindness associated with DR and VTDR

The proportions of visual impairment and blindness were significantly higher in the participants with any DR than those without DR, and in the participants with VTDR than in those with non-VTDR (all $p < 0.050$). The rates of worse-seeing and better-seeing eye blindness among the patients with VTDR were 11.25-fold (95% CI 8.13–15.58) and 10.26-fold (95% CI 5.97–17.65) higher than those with non-VTDR, respectively, after adjustment for sex and age (Table 5).

### Factors associated with prevalent DR, non-VTDR, and VTDR

Multivariable-adjusted analyses results assessing the factors associated with any DR and the severity of DR were shown in Table 6 (for demographic and clinical factors) and Table 7 (for lifestyle factors).

Females were less likely to have any DR, or non-VTDR and VTDR than males, with odds ratio (OR) (95% confidence interval [CI]) being 0.78 (0.69–0.87), 0.82 (0.72–0.92), and 0.63 (0.53–0.75), respectively. Age was only significantly negatively associated with VTDR (OR 0.98, 95% CI 0.97–0.99). The people with diabetes living in the northern region were more likely to have any DR (OR 1.39, 95% CI 1.22–1.58), non-VTDR (OR 1.35, 95% CI 1.17–1.55) and VTDR (OR 1.60, 95% CI 1.34–1.91) than those in the southern regions (Table 6).

As for socio-economic indicators, the lower likelihoods of having any DR, non-VTDR, and VTDR were found in the participants with education levels of high school and above, with ORs ranging from 0.77 (95% CI 0.61–0.96) to 0.84 (95% CI 0.74–0.95), and average annual household income per capita equal to or greater than 20,000 RMB, with ORs ranging from 0.65 (95% CI 0.53–0.81) to 0.79 (95% CI 0.69–0.91), than their counterparts (Table 6).

In terms of clinical characteristics, the participants with longer diabetes duration and family histories of diabetes showed significantly higher odds of any DR, non-VTDR, or VTDR (both $p < 0.050$). The participants who exhibited poor HbA1c or BP control had 2.25-fold or 1.30-fold higher likelihoods of having any DR, 2.17-fold or 1.29-fold higher odds of having non-VTDR, and 2.72-fold or 1.33-fold higher risks

**Table 1 | Characteristics of study participants by the presence of any DR**

| Characteristics | Total (n = 50,564) | Any DR | | |
|---|---|---|---|---|
| | | No DR (n = 42,005) | Any DR (n = 8559) | p[a] |
| Demographics | | | | |
| Female | 25,448 (50.3) | 21,284 (50.7) | 4164 (48.7) | 0.0007 |
| Age, y | 57.5 (50.9–64.8) | 57.4 (50.7–64.9) | 57.8 (51.8–64.7) | 0.0005 |
| Geographic region | | | | |
| North[b] | 24,862 (49.2) | 20,106 (47.9) | 4756 (55.6) | <0.0001 |
| Setting | | | | |
| Rural | 23,639 (46.8) | 19,427 (46.2) | 4212 (49.2) | <0.0001 |
| Socio-economic status | | | | |
| High school and above | 12,946 (25.6) | 11,089 (26.4) | 1857 (21.7) | <0.0001 |
| Average annual household income per capita | | | | <0.0001 |
| <¥10,000 | 14,507 (28.7) | 11,769 (28.0) | 2738 (32.0) | |
| ¥10,000- <¥20,000 | 10,049 (19.9) | 8317 (19.8) | 1732 (20.2) | |
| ≥¥20,000 | 13,877 (27.4) | 11,878 (28.3) | 1999 (23.4) | |
| Unwilling to disclose | 12,131 (24.0) | 10,041 (23.9) | 2090 (24.4) | |
| Clinical characteristics[c] | | | | |
| History of DR | 3033 (6.0) | 1844 (4.4) | 1189 (13.6) | <0.0001 |
| Diabetes duration, y | 5.2 (2.5–10.1) | 4.9 (2.3–9.2) | 8.4 (4.2–13.6) | <0.0001 |
| Family history of diabetes | 20,766 (41.1) | 16,901 (40.2) | 3865 (45.2) | <0.0001 |
| BMI, kg/m² | 25.5 (23.3–27.8) | 25.5 (23.3–27.9) | 25.3 (23.1–27.7) | <0.0001 |
| FPG, mmol/L | 8.06 (6.55–10.41) | 7.85 (6.43–9.96) | 9.48 (7.48–12.54) | <0.0001 |
| HbA1c, % | 7.2 (6.2–8.6) | 7.0 (6.1–8.4) | 8.2 (7.0–9.8) | <0.0001 |
| SBP, mmHg | 135.3 (123.0–149.3) | 134.3 (122.7–148.0) | 139.3 (126.0–154.3) | <0.0001 |
| DBP, mmHg | 79.7 (72.7–87.0) | 79.7 (72.7–87.0) | 80.0 (72.3–87.7) | 0.0268 |
| HDL-C, mmol/L | 1.21 (1.01–1.46) | 1.21 (1.01–1.46) | 1.22 (1.01–1.47) | 0.1129 |
| LDL-C, mmol/L | 2.96 (2.33–3.61) | 2.95 (2.32–3.59) | 3.00 (2.37–3.69) | <0.0001 |
| TG, mmol/L | 1.76 (1.22–2.66) | 1.77 (1.23–2.67) | 1.73 (1.20–2.64) | 0.0198 |
| Medications[d] | | | | |
| Glucose-lowering treatment | 39,733 (78.6) | 32,313 (76.9) | 7420 (86.7) | <0.0001 |
| Antihypertensive treatment | 19,755 (39.1) | 16,331 (38.9) | 3424 (40.0) | 0.0519 |
| Lipid-lowering treatment | 6236 (12.3) | 5214 (12.4) | 1022 (11.9) | 0.2257 |
| Attainment of targets | | | | |
| HbA1c < 7.0 % | 22,229 (44.0) | 20,156 (48.0) | 2073 (24.2) | <0.0001 |
| BP <130/80 mmHg | 15,044 (29.8) | 12,842 (30.6) | 2202 (25.7) | <0.0001 |
| LDL-C < 2.6 mmol/L | 17,628 (34.9) | 14,790 (35.3) | 2838 (33.2) | 0.0003 |
| Lifestyle factors | | | | |
| Physical activity ≥600 MET minutes/week[e] | 39,777 (78.7) | 33,243 (79.1) | 6534 (76.3) | <0.0001 |
| Current smoker | 11,698 (23.1) | 9787 (23.3) | 1911 (22.3) | 0.0519 |
| Current drinker | 13,811 (27.3) | 11,628 (27.7) | 2183 (25.5) | <0.0001 |
| Dietary intake | | | | |
| Refined grains, g/day | 300 (160–450) | 300 (150–450) | 300 (160–450) | 0.1961 |
| Whole grains, g/day | 14.3 (1.7–50.0) | 14.3 (1.7–50.0) | 14.3 (1.7–50.0) | 0.0732 |
| Potatoes, g/day | 11.4 (1.7–30.8) | 12.9 (1.7–34.3) | 8.6 (0.5–28.6) | <0.0001 |
| Soybean products, g/day | 14.3 (3.3–30.0) | 14.3 (3.3–30.0) | 14.3 (3.3–28.6) | 0.0033 |
| Fresh vegetables, g/day | 300 (150–500) | 300 (150–500) | 300 (150–500) | 0.0204 |
| Fresh fruits, g/day | 28.6 (3.3–100.0) | 28.6 (3.3–100.0) | 21.4 (0.3–77.1) | <0.0001 |
| Dairy products, ml/day | 1.4 (0.0–100.0) | 1.4 (0.0–100.0) | 0.0 (0.0–85.7) | 0.1388 |
| Red meat, g/day | 42.9 (14.3–100.0) | 42.9 (14.3–100.0) | 40.0 (12.9–100.0) | <0.0001 |
| Poultry, g/day | 6.7 (0.9–15.0) | 6.7 (1.0–15.7) | 5.0 (0.7–14.3) | <0.0001 |
| Seafood, g/day | 6.7 (0.8–25.7) | 7.1 (0.8–28.6) | 6.7 (0.4–21.4) | <0.0001 |
| Eggs, g/day | 25.7 (7.9–50.0) | 25.7 (7.9–50.0) | 25.0 (7.9–50.0) | 0.5227 |
| Nuts >0 g/day | 34,309 (67.9) | 28,741 (68.4) | 5568 (65.1) | <0.0001 |
| Fresh juices >0 ml/day | 3464 (6.9) | 2977 (7.1) | 487 (5.7) | <0.0001 |

Any *DR* any diabetic retinopathy. *BM* body-mass index. *FPG* fasting plasma glucose. *HbA1c* hemoglobin A1c. *SBP* systolic blood pressure. *DBP* diastolic blood pressure. *HDL-C* high-density lipoprotein cholesterol. *LDL-C* low-density lipoprotein cholesterol. *TG* triglycerides. *BP* blood pressure. *MET* metabolic equivalent.

Data were presented as median (25th percentile–75th percentile) for continuous variables or number (percentage) for categorical variables.

[a]*p* value was calculated using the two-sided Wilcoxon rank test for continuous variables, and the two-sided Chi-Square test for categorical variables.

[b]The Northern region includes Beijing, Tianjin, Hebei, Shanxi, Inner Mongolia, Liaoning, Jilin, Heilongjiang, Shandong, Henan, Shaanxi, Gansu, Qinghai, Ningxia, and Xinjiang; the Southern region includes Shanghai, Jiangsu, Zhejiang, Anhui, Fujian, Jiangxi, Hubei, Hunan, Guangdong, Guangxi, Hainan, Chongqing, Sichuan, Guizhou, Yunnan, and Tibet.

[c]There were 45, 27, 117, 117, and 117 missing values for FPG, HbA1c, HDL-C, LDL-C, and TG, respectively. In addition, there were 196 missing values for history of DR, 12 for body-mass index, and 30 for blood pressure.

[d]Glucose-lowering treatment included oral agent therapy and/or insulin therapy. Antihypertensive treatment included angiotensin-converting enzyme inhibitor, angiotensin receptor blocker, aldosterone, β-blocker, α-blocker, diuretic, calcium antagonist, and others. Lipid-lowering treatment included statin, fibrate, and others.

[e]MET was calculated according to a total of moderate- and vigorous-intensity physical activity (moderate MET value was equal to 4.0, and vigorous MET value was equal to 8.0) for work, in-transit, and leisure time throughout a week.

Source data are provided as a Source Data file.

**Table 2 | Weighted prevalence of any DR and VTDR among Chinese adults with diabetes by demographic factors and diabetes duration**

| Subpopulation | N | Any DR | | | VTDR | | | |
|---|---|---|---|---|---|---|---|---|
| | | Total | DR | DME | Total | Severe NPDR | PDR | CSME |
| Weighted number | 119,749,193 | 19,466,938 | 19,407,184 | 893,750 | 3,814,283 | 2,783,391 | 765,756 | 646,935 |
| Number | 50,564 | 8559 | 8529 | 431 | 1673 | 1245 | 295 | 311 |
| Total | 50,564 | 16.3 (15.3–17.2) | 16.2 (15.3–17.1) | 0.75 (0.64–0.86) | 3.2 (2.9–3.5) | 2.3 (2.1–2.6) | 0.64 (0.55–0.73) | 0.54 (0.45–0.63) |
| Sex | | | | | | | | |
| Male | 25,116 | 16.6 (15.7–17.6) | 16.6 (15.6–17.5) | 0.72 (0.60–0.85) | 3.2 (2.9–3.5) | 2.5 (2.2–2.7) | 0.59 (0.49–0.69) | 0.53 (0.44–0.63) |
| Female | 25,448 | 15.8 (14.6–17.0) | 15.8 (14.6–17.0) | 0.77 (0.64–0.90) | 3.1 (2.7–3.5) | 2.2 (1.9–2.5) | 0.70 (0.52–0.88) | 0.55 (0.44–0.66) |
| p for difference[a] | | 0.1449 | | | 0.6856 | | | |
| Age | | | | | | | | |
| 18- < 45 y | 7031 | 12.9 (11.1–14.6) | 12.8 (11.1–14.6) | 0.39 (0.27–0.52) | 2.7 (2.0–3.3) | 1.7 (1.2–2.2) | 0.89 (0.59–1.18) | 0.25 (0.15–0.35) |
| 45- < 60 y | 22,880 | 18.0 (16.8–19.2) | 18.0 (16.8–19.1) | 0.89 (0.72–1.06) | 3.6 (3.3–3.9) | 2.7 (2.4–3.0) | 0.62 (0.52–0.73) | 0.69 (0.55–0.84) |
| ≥60 y | 20,653 | 17.3 (16.2–18.3) | 17.2 (16.2–18.2) | 0.90 (0.74–1.05) | 3.2 (2.8–3.5) | 2.4 (2.2–2.7) | 0.43 (0.35–0.52) | 0.63 (0.49–0.76) |
| p for linear trend[b] | | <0.0001 | | | 0.2081 | | | |
| Geographical region[c] | | | | | | | | |
| South | 25,702 | 14.4 (13.3–15.5) | 14.3 (13.2–15.4) | 0.57 (0.41–0.73) | 2.5 (2.2–2.8) | 1.8 (1.6–2.1) | 0.52 (0.42–0.62) | 0.39 (0.27–0.51) |
| North | 24,862 | 18.1 (16.6–19.6) | 18.0 (16.5–19.5) | 0.92 (0.77–1.07) | 3.8 (3.4–4.3) | 2.8 (2.4–3.2) | 0.76 (0.61–0.90) | 0.68 (0.57–0.80) |
| p for difference[a] | | 0.0001 | | | <0.0001 | | | |
| Setting | | | | | | | | |
| Rural | 23,639 | 16.9 (15.4–18.3) | 16.8 (15.3–18.3) | 0.76 (0.57–0.95) | 3.2 (2.8–3.6) | 2.3 (2.0–2.7) | 0.60 (0.51–0.70) | 0.57 (0.42–0.71) |
| Urban | 26,925 | 15.5 (13.8–17.2) | 15.4 (13.8–17.1) | 0.73 (0.58–0.88) | 3.2 (2.7–3.7) | 2.3 (1.9–2.8) | 0.69 (0.52–0.85) | 0.51 (0.40–0.62) |
| p for difference[a] | | 0.2933 | | | 0.9231 | | | |
| Diabetes duration | | | | | | | | |
| <1 y | 4610 | 8.2 (6.7–9.7) | 8.1 (6.6–9.6) | 0.25 (0.11–0.38) | 1.24 (0.49–2.00) | 0.99 (0.26–1.72) | 0.24 (0.09–0.38) | 0.17 (0.05–0.30) |
| 1- < 10 y | 33,003 | 13.5 (12.6–14.4) | 13.4 (12.5–14.4) | 0.54 (0.44–0.63) | 2.1 (1.8–2.4) | 1.5 (1.3–1.8) | 0.45 (0.32–0.58) | 0.38 (0.30–0.45) |
| 10- < 20 y | 11,143 | 27.1 (25.4–28.9) | 27.1 (25.4–28.8) | 1.6 (1.2–1.9) | 6.3 (5.6–7.0) | 4.7 (4.1–5.3) | 1.18 (0.97–1.40) | 1.15 (0.91–1.39) |
| ≥20 y | 1808 | 38.6 (35.8–41.4) | 38.5 (35.7–41.3) | 2.1 (1.5–2.7) | 13.8 (11.9–15.6) | 10.0 (8.3–11.7) | 2.9 (2.2–3.7) | 1.6 (1.1–2.1) |
| p for linear trend[b] | | <0.0001 | | | <0.0001 | | | |

Any *DR* any diabetic retinopathy. *DME* diabetic macular edema. *VTDR* vision-threatening diabetic retinopathy. *NPDR* non-proliferative diabetic retinopathy. *PDR* proliferative diabetic retinopathy.
*CSME* clinically significant macular edema. *HbA1c* hemoglobin A1c. *BP* blood pressure. *LDL-C* low-density lipoprotein cholesterol.
Data were presented as weighted percentage (95% confidence interval), which were weighted by the sex-, age-, and rural/urban structure of adults with diabetes aged 18–74 years in China in 2018 from the China Chronic Disease and Risk Factors Surveillance system.
[a]p for difference was calculated using the two-sided Rao-Scott Chi-Square test.
[b]p for linear trend was evaluated using a logistic regression model (for two-sided test) with median values of each category to represent their levels.
[c]The Northern region includes Beijing, Tianjin, Hebei, Shanxi, Inner Mongolia, Liaoning, Jilin, Heilongjiang, Shandong, Henan, Shaanxi, Gansu, Qinghai, Ningxia, and Xinjiang; the Southern region includes Shanghai, Jiangsu, Zhejiang, Anhui, Fujian, Jiangxi, Hubei, Hunan, Guangdong, Guangxi, Hainan, Chongqing, Sichuan, Guizhou, Yunnan, and Tibet.
Source data are provided as a Source Data file.

of having VTDR, respectively, than those with better controls. Nevertheless, there were no significant associations of LDL-C control with any DR, non-VTDR, or VTDR (Table 6).

Further, analyses on the association of physical activity and diet with any DR, non-VTDR, and VTDR found that physical activity over 600 metabolic equivalents (METs) minutes/week were significantly negatively associated with any DR (OR 0.88, 95% CI 0.77–0.99). In terms of diet, higher fresh fruit consumption (>100 g/day) was negatively associated with any DR (OR 0.85, 95% CI 0.77–0.95), as well as non-VTDR and VTDR with ORs of 0.87 (95% CI 0.77–0.99) and 0.77 (95% CI 0.63–0.95). Additionally, higher potato intake (>31 g/day) was negatively associated with any DR (OR 0.86, 95% CI 0.79–0.94) and non-VTDR (OR 0.84, 95% CI 0.78–0.91), and higher dairy product consumption (>100 ml/day) was only negatively associated with non-VTDR (OR 0.90, 95% CI 0.84–0.98). In contrast, higher refined grain consumption (>450 g/day) was positively associated with non-VTDR (OR 1.11, 95% CI 1.01–1.22) (Table 7).

In addition, a comparison of characteristics of Chinese adults with diabetes in the northern and southern regions was shown in Supplementary Table 2.

## Discussion

Our study was the first to report a nationally representative prevalence of 16.3% (15.3%–17.2%) for any DR (16.2% [15.3%–17.1%] for DR and 0.75% [0.64%–0.86%] for DME) and 3.2% (2.9%–3.5%) for VTDR (2.3% [2.1%–2.6%] for severe NPDR, 0.64% [0.55%–0.73%] for PDR, and 0.54% [0.45%–0.63%] for CSME) in Chinese adults with diagnosed diabetes aged 18–74 years. The previous prevalence rates of DR derived from different studies were only used as rough references because there were differences in study methodologies and some details of individual studies were not provided. By pooling data from the population-based studies among adults with diabetes aged 20 and older, the global pooled prevalence, of 27 countries between 1980 and 2017, were 22.27% for any DR and 6.17% for VTDR[12], and the national pooled

**Table 3 | Weighted prevalence of any DR and VTDR among Chinese adults with diabetes by attainment of metabolic targets**

| Subpopulation | N | Any DR | | | VTDR | | | |
|---|---|---|---|---|---|---|---|---|
| | | Total | DR | DME | Total | Severe NPDR | PDR | CSME |
| HbA1c | | | | | | | | |
| <7.0 % | 22,229 | 8.7 (7.8–9.6) | 8.7 (7.8–9.5) | 0.30 (0.23–0.38) | 1.4 (1.1–1.6) | 0.97 (0.75–1.20) | 0.30 (0.21–0.38) | 0.18 (0.13–0.24) |
| ≥7.0 % | 28,308 | 22.3 (21.1–23.4) | 22.2 (21.1–23.4) | 1.10 (0.92–1.28) | 4.6 (4.2–5.0) | 3.4 (3.0–3.8) | 0.91 (0.77–1.06) | 0.83 (0.69–0.96) |
| p for difference[a] | | <0.0001 | | | <0.0001 | | | |
| BP | | | | | | | | |
| <130/80 mmHg | 15,044 | 13.7 (12.5–14.9) | 13.6 (12.4–14.9) | 0.54 (0.41–0.68) | 2.7 (2.3–3.1) | 1.9 (1.6–2.2) | 0.63 (0.41–0.85) | 0.40 (0.30–0.51) |
| ≥130/80 mmHg | 35,490 | 17.5 (16.5–18.5) | 17.5 (16.4–18.5) | 0.85 (0.72–0.97) | 3.4 (3.1–3.7) | 2.5 (2.2–2.8) | 0.64 (0.55–0.73) | 0.61 (0.50–0.71) |
| p for difference[a] | | <0.0001 | | | 0.0085 | | | |
| LDL-C | | | | | | | | |
| <2.6 mmol/L | 17,628 | 15.4 (14.3–16.5) | 15.4 (14.3–16.5) | 0.56 (0.43–0.68) | 2.9 (2.5–3.3) | 2.2 (1.9–2.6) | 0.52 (0.37–0.67) | 0.39 (0.28–0.50) |
| ≥2.6 mmol/L | 32,819 | 16.7 (15.7–17.8) | 16.7 (15.6–17.8) | 0.86 (0.71–1.00) | 3.4 (3.1–3.7) | 2.4 (2.1–2.7) | 0.71 (0.59–0.83) | 0.63 (0.51–0.74) |
| p for difference[a] | | 0.0287 | | | 0.0641 | | | |

Any DR any diabetic retinopathy. DME diabetic macular edema. VTDR vision–threatening diabetic retinopathy. NPDR non-proliferative diabetic retinopathy. PDR proliferative diabetic retinopathy. CSME clinically significant macular edema. HbA1c hemoglobin A1c. BP blood pressure. LDL-C low-density lipoprotein cholesterol.
Data were presented as weighted percentage (95% confidence interval), which were weighted by the sex-, age-, and rural/urban structure of adults with diabetes aged 18–74 years in China in 2018 from the China Chronic Disease and Risk Factors Surveillance system. Data were presented as weighted prevalence (95% confidence interval).
[a]p for difference was calculated using the two-sided Rao-Scott Chi-Square test.
Source data are provided as a Source Data file.

prevalence for China, of 16 provinces between 1990 and 2017, were 18.45% for any DR and 0.99% for PDR[13]. A lower prevalence of any DR in China may be partly related to ethnic disparity, with Asians reported having a lower DR prevalence than the Hispanics[12], Middle Easterners[12], and Caucasians[1], and that the diabetes epidemic commenced later in China than in some developed countries[14,15].

Previous small-sample size local population-based surveys in Chinese with diabetes reported a significant discrepancy in the prevalence of DR with a range of 5.4%–41.7%[13], which not only reflects differences in region-related factors, but is also probably due to different study designs, grading standards, and population characteristics. Based on the same methodologies, central blind grading, and unified quality control, our study reported national prevalence rates of DR and PDR slightly lower than the pooled rates for Chinese patients (16.3% vs. 18.45% for DR; 0.64% vs. 0.99% for PDR), and exhibited variations in province-specific prevalence of any DR from 9.9% to 29.1% and VTDR from 1.27% to 6.3% among Chinese adults with diabetes aged 18–74 years. This study showed that the top three province-specific prevalence of any DR and VTDR were all in the northern regions, and generally, a higher prevalence of any DR and VTDR was observed in the northern regions than in the southern regions. The north–south variation was in line with two meta-analysis studies regarding the regional distribution of DR in China[13,16]. Higher DR prevalence in rural than in urban areas reported by the two meta-analyses, was not seen in this study[13,16]. In general, the prevalence of DR in this study seemed slightly lower than that in the previous reports, which may be attributed to improved diabetes care and expanded screening coverage for DR to some extent. In addition, there seemed to be a decreasing trend in the prevalence of DR over time in China when comparing the prevalence rates between the two meta-analyses (from 23.0% between 1986 and 2009 to 18.45% between 1990 and 2017) and between the two multi-province surveys with participants from community health service centers and hospitals (from 34.08% between 2014 and 2015 to 30.1% between 2015 and 2018)[8,9,13,16].

Some studies observed that DR was also found in patients with newly diagnosed diabetes, for example, 18% in patients with newly diagnosed T2DM in England[17]. Our study showed that the prevalence of any DR, especially VTDR and its subtypes, significantly increased with prolonged diabetes duration: the prevalence of any DR and VTDR reached 8.2% (6.7%–9.7%) and 1.24% (0.49%–2.00%) with diabetes duration of less than one year, and rose to 38.6% (35.8%–41.4%) and

13.8% (11.9%–15.6%) with diabetes duration of 20 years or more, respectively. A study showed that two years after the diagnosis of DR, the probabilities of sustained blindness in eyes with moderate NPDR, severe NPDR, and PDR were 2.6, 3.6, and 4.0 times higher than in eyes with mild NPDR, respectively[18]. In line with these results, our study observed that the proportions of worse-seeing and better-seeing eye blindness were 11.25-fold (95% CI 8.13–15.58) and 10.26-fold (95% CI 5.97–17.65) higher for patients with VTDR than those with non-VTDR. Regular screening for DR recommended by the American Diabetes Association and Chinese Diabetes Society and the establishment of a comprehensive eye screening system in China are necessary strategies to decrease vision loss caused by DR.

Consistent with the established knowledge, DR and VTDR were more prevalent among patients with worse control of glycemia and BP[6]. In this study, less than half of the participants achieved the recommended levels of HbA1c, BP, and LDL-C (44.0%, 29.8%, and 34.9%), lower than in US adults with diabetes between 1999 and 2018 (51.1%, 47.0%, and 53.3%)[14]. Hence, there is still much room for improvement in the metabolic management of patients with diabetes in China.

Our study showed that people with a family history of diabetes or living in northern China were more likely to have DR than their counterparts even after adjustment for multiple factors. Genetic background and shared environmental factors contributed to the susceptibility to DR. A family clustering study showed that genetic components seem to contribute more to the severity of DR than to the presence of DR[19]. In China, the Qinling–Huaihe line is the most commonly used line to divide the northern or southern regions. The two regions not only have significant differences in natural conditions and socio-cultural customs[20], but also differences in physical features[21] and genetic background[22]. Further exploration of the causes of geographical differences may provide more clues into possible genetic and environmental contributors to the etiology of DR.

Several socio-economic factors were also involved in the development of DR. Our study showed that participants educated to high school and above were less likely to have any DR, which emphasized the importance of improving the knowledge of diabetic prevention at the population levels. As a Canadian cohort study showed, low-income people were less likely to engage in preventive care and tended to have a higher prevalence and greater severity of DR[23]. Screening for DR is cost-effective, but it needs to address some barriers, such as

**Table 4 | Standardized province-specific prevalence of any DR and VTDR among Chinese adults with diabetes**

| Province | N | Any DR | | VTDR | |
|---|---|---|---|---|---|
| | | Prevalence (95% CI)[a] | Rank | Prevalence (95% CI)[a] | Rank |
| Guizhou | 1727 | 9.9 (8.3–11.5) | 1 | 1.8 (1.2–2.4) | 3 |
| Fujian | 1744 | 10.9 (9.3–12.5) | 2 | 2.2 (1.5–2.9) | 6 |
| Tianjin | 1729 | 11.2 (9.5–12.9) | 3 | 2.7 (1.9–3.5) | 12 |
| Qinghai | 802 | 11.5 (9.0–13.9) | 4 | 3.7 (2.2–5.3) | 23 |
| Guangxi | 1584 | 11.5 (9.6–13.4) | 5 | 1.64 (0.81–2.46) | 2 |
| Jiangsu | 1768 | 11.6 (9.9–13.4) | 6 | 2.6 (1.8–3.4) | 11 |
| Hubei | 1607 | 12.0 (10.0–14.1) | 7 | 1.9 (1.1–2.7) | 5 |
| Shanxi | 1858 | 12.4 (10.7–14.0) | 8 | 2.8 (2.0–3.6) | 13 |
| Hunan | 1676 | 12.5 (10.6–14.4) | 9 | 2.5 (1.4–3.5) | 9 |
| Shanghai | 1723 | 12.6 (10.9–14.4) | 10 | 2.5 (1.7–3.3) | 10 |
| Jiangxi | 1453 | 14.4 (11.9–16.8) | 11 | 1.27 (0.61–1.94) | 1 |
| Ningxia | 1719 | 15.1 (13.1–17.1) | 12 | 1.9 (1.2–2.5) | 4 |
| Tibet | 578 | 15.2 (11.8–18.7) | 13 | 3.7 (2.3–5.2) | 24 |
| Liaoning | 1701 | 15.5 (13.5–17.5) | 14 | 3.7 (2.7–4.8) | 22 |
| Hebei | 1790 | 15.6 (13.6–17.6) | 15 | 3.4 (2.4–4.5) | 18 |
| Anhui | 1636 | 15.7 (13.5–17.8) | 16 | 2.4 (1.6–3.2) | 7 |
| Zhejiang | 1639 | 16.0 (13.9–18.1) | 17 | 3.5 (2.5–4.5) | 19 |
| Guangdong | 1570 | 16.1 (13.8–18.3) | 18 | 3.0 (2.0–4.0) | 14 |
| Inner Mongolia | 1633 | 16.5 (14.4–18.6) | 19 | 2.4 (1.7–3.2) | 8 |
| Yunnan | 1719 | 16.8 (14.7–18.9) | 20 | 3.8 (2.7–4.9) | 25 |
| Sichuan | 2145 | 17.4 (15.5–19.4) | 21 | 3.9 (3.1–4.8) | 26 |
| Jilin | 1568 | 17.9 (15.6–20.2) | 22 | 4.5 (3.3–5.6) | 28 |
| Shaanxi | 1698 | 18.1 (15.9–20.2) | 23 | 4.0 (3.0–5.0) | 27 |
| Beijing | 1708 | 18.6 (16.3–20.9) | 24 | 4.7 (3.6–5.9) | 29 |
| Hainan | 1462 | 18.7 (16.2–21.2) | 25 | 3.1 (2.0–4.2) | 15 |
| Gansu | 1710 | 18.7 (16.5–20.9) | 26 | 3.7 (2.7–4.6) | 21 |
| Xinjiang | 856 | 19.4 (16.4–22.5) | 27 | 3.4 (2.2–4.6) | 17 |
| Chongqing | 1671 | 19.8 (17.4–22.1) | 28 | 3.1 (2.1–4.1) | 16 |
| Henan | 2162 | 24.6 (22.3–26.8) | 29 | 3.6 (2.8–4.5) | 20 |
| Heilongjiang | 1761 | 27.0 (24.5–29.5) | 30 | 6.3 (5.1–7.5) | 31 |
| Shandong | 2167 | 29.1 (26.5–31.6) | 31 | 6.2 (5.0–7.3) | 30 |

*Any DR* any diabetic retinopathy. *VTDR* vision-threatening diabetic retinopathy. *CI* confidence interval.

[a]Data were presented as weighted percentage (95% confidence interval), which were direct standardized by the sex- and age- structure of adults with diabetes aged 18–74 years in China in 2018 from the China Chronic Disease and Risk Factors Surveillance system.

Source data are provided as a Source Data file.

acceptability, availability, and affordability. Artificial intelligence diagnostic systems are expected to offer a promising solution to this dilemma in the future[24]. Unfortunately, our study showed that the majority of people with DR (86.5% of the participants with any DR, 90.6% of participants with non-VTDR, and 67.2% of participants with VTDR) were undiagnosed before this survey.

Our findings also suggested that patients without DR had healthier dietary patterns than those without DR, with higher intakes of fresh fruits, potatoes, and dairy products, but a lower intake of refined grains. In particular, fresh fruit intakes were favorably associated with any DR and VTDR. Fresh fruits are an important source of essential vitamins, minerals, fiber, and flavonoids that can help decrease retinal injury[25]. However, in this study, the patients' fruit intake was well below the recommended intake of the Dietary Guidelines for Chinese Residents (28.6 g/day vs 200.0 g/day). Therefore, healthy diet guidance for patients with diabetes needs to be on the agenda in China.

Our study has several strengths. Firstly, it was the national, population-based survey of DR with a multistage sampling scheme. Together with a systematic and comprehensive investigation of the associated risk factors, including not only information on socio-demographics, medical history, and clinical data, but also detailed information on lifestyle, it is then possible to describe multi-level factors associated with DR. Secondly, professional ophthalmologists checked the fundus photographs with unified criteria on one center. Several limitations require consideration. Firstly, the two-field fundus photography was used instead of optical coherence tomography, which may affect the accurate classification of DME. Also, this study is conducted in community health centers instead of being completed in specialized ophthalmology departments in hospitals. Due to limited resources, it is difficult to include the assessment of some ocular risk parameters for DR, like hyperopia or short axial length, in a large-scale epidemiology study. Secondly, a temporal relationship between exposure and outcome cannot be confirmed, and there were inevitable misreporting and recall biases. Thirdly, the differences between those with ungradable photos and those with gradable photos might introduce selection bias. But due to the very low proportion of ungradable photos (3.01%), the effect was minimal.

In conclusion, in China, approximately 19.5 million people with diabetes had any DR; of them, one-fifth are at the VTDR stage. With a large number of people with diabetes and an aging population in China comes the great challenge of avoiding visual impairment and blindness. Our study showed that multifaceted and tailored efforts to reduce the vision loss of patients with diabetes, including early and regular screening for DR, metabolic control improvement, educational improvement, lifestyle promotion, more care for these vulnerable and high-risk populations, and further exploration of geographical causes are necessary.

## Methods

The study protocol was approved by the Ethics Committee of Shanghai Sixth People's Hospital (Approval No: 2018-010) and was also registered in the Chinese Clinical Trial Registry (ChiCTR1800014432). All the study participants provided written informed consent before data collection.

The study protocol was published before[26] and summarized briefly below.

### Study design and study participants

The China National Diabetic Chronic Complications Study (China DiaChronic Study) was conducted to investigate the epidemiological characteristics of diabetes-related complications and the attainment rates of metabolic targets in adults with diagnosed diabetes in China between March 2018 and January 2020. All those recruited in this study were people with diabetes diagnosed by physicians in hospitals, registered in the diabetes management registration system of basic public health services[27] in community health centers, and monitored by the local Center for Disease Control and Prevention. A multistage sampling scheme (stratification, clustering, and random selection) was designed based on the disease surveillance points of the China Chronic Disease and Risk Factors Surveillance (CCDRFS)[28]. 58560 participants aged 18–74 years were sampled from the diabetes management registration system of 488 neighborhoods/villages across 31 provinces, autonomous regions, and municipalities (referred to as provinces hereafter). A flowchart of the multistage sampling scheme was listed in the protocol of this study[26]. Briefly speaking, there are three sampling stages. In the first stage, four study sites based on the disease surveillance points or replaced study sites were selected from each province after considering urbanization levels. Finally, a total of 122 study sites (65 urban study sites and 57 rural study sites) were

**Table 5 | Sex- and age-adjusted odds ratios of distant visual impairment associated with DR and VTDR**

| Dependent variables | Total | Any DR | | | | Any DR | | | |
|---|---|---|---|---|---|---|---|---|---|
| | | No DR | Any DR | OR (95% CI)[a] | p[a] | Non-VTDR | VTDR | OR (95% CI)[a] | p[a] |
| Worse-seeing eye[b] | | | | | | | | | |
| N | 49,600 | 41,261 | 8339 | | | 6744 | 1595 | | |
| Normal | 34,058 (68.7) | 29,085 (70.5) | 4973 (59.6) | 1 (ref) | | 4367 (64.8) | 606 (38.0) | 1 (ref) | |
| Mild | 4641 (9.4) | 3748 (9.1) | 893 (10.7) | 1.56 (1.33–1.84) | <0.0001 | 737 (10.9) | 156 (9.8) | 1.52 (1.08–2.14) | 0.0176 |
| Moderate | 9472 (19.1) | 7459 (18.1) | 2013 (24.1) | 1.64 (1.46–1.83) | <0.0001 | 1420 (21.1) | 593 (37.2) | 2.83 (2.36–3.38) | <0.0001 |
| Severe | 453 (0.91) | 325 (0.79) | 128 (1.5) | 3.09 (1.68–5.70) | 0.0003 | 79 (1.2) | 49 (3.1) | 2.75 (1.13–6.72) | 0.0269 |
| Blindness | 976 (2.0) | 644 (1.6) | 332 (4.0) | 3.08 (2.37–4.00) | <0.0001 | 141 (2.1) | 191 (12.0) | 11.25 (8.13–15.58) | <0.0001 |
| Better-seeing eye[c] | | | | | | | | | |
| N | 50,236 | 41,739 | 8497 | | | 6838 | 1659 | | |
| Normal | 41,747 (83.1) | 35,241 (84.4) | 6506 (76.6) | 1 (ref) | | 5510 (80.6) | 996 (60.0) | 1 (ref) | |
| Mild | 3000 (6.0) | 2374 (5.7) | 626 (7.4) | 1.58 (1.27–1.97) | <0.0001 | 485 (7.1) | 141 (8.5) | 1.42 (1.02–1.97) | 0.0367 |
| Moderate | 5073 (10.1) | 3846 (9.2) | 1227 (14.4) | 1.84 (1.59–2.12) | <0.0001 | 786 (11.5) | 441 (26.6) | 2.73 (2.25–3.32) | <0.0001 |
| Severe | 150 (0.30) | 109 (0.26) | 41 (0.48) | 4.60 (1.50–14.09) | 0.0067 | 19 (0.28) | 22 (1.3) | 2.02 (0.53–7.78) | 0.3021 |
| Blindness | 266 (0.53) | 169 (0.40) | 97 (1.1) | 3.07 (1.96–4.83) | <0.0001 | 38 (0.56) | 59 (3.6) | 10.26 (5.97–17.65) | <0.0001 |

*DR* diabetic retinopathy. *VTDR* vision-threatening diabetic retinopathy. *OR* dds ratio. *CI* confidence interval.

Data were presented as number (percentage) unless otherwise stated.

[a]The multinomial surveylogistic regression (for two-sided test) was used to calculate OR (95% CI) and *p* value, where the severity of the distant visual impairment (normal, mild, moderate, severe, and blindness; normal as the referent category) was treated as a dependent variable, and the presence or absence of any DR or VTDR was treated as the independent variable (presence vs absence), respectively, after adjusting for age and gender.

[b]Data were analyzed after excluding participants with other cause-related blindness (N = 729), including cataracts, eye trauma, high myopia, keratopathy (keratitis, corneal degeneration, and corneal dystrophy), retinopathy (macular degeneration, retinal detachment), optic neuropathy, choroidopathy, glaucoma, strabismus, vitreous diseases (vitreous opacity, vitreous hemorrhage), nystagmus, presbyopia, ocular tumors, pterygium, amblyopia, intraocular lens dislocation, congenital and hereditary eye diseases, measles sequela, and other diseases (cerebral infarction, sequela of cerebral infarction).

[c]Data were analyzed after excluding participants with other cause-related blindness (N = 93), including high myopia, retinopathy (macular degeneration, retinal detachment), cataract, eye trauma, congenital and hereditary eye diseases, glaucoma, presbyopia, keratopathy (corneal degeneration), nystagmus, ocular tumors, vitreous diseases (vitreous opacity), choroidopathy, and other diseases (sequela of cerebral infarction).

Source data are provided as a Source Data file.

randomly selected and invited to participate. In the second stage, four neighborhoods in urban areas or four villages in rural areas were randomly selected from each study site, resulting in 260 neighborhoods and 228 villages in total. In the third stage, according to the age and gender structure of the CCDRFS 2013 diabetes data, the national sample size of 58,560 individuals and the sample size of 480 at each study site were set. 480 participants were randomly invited from those registered in the diabetes management registration system at each study site.

53,401 participants completed the interview with an overall response rate of 91.2%. Retinal photographs were not taken in 1267 participants and were of insufficient quality for grading in 1570 participants. The comparisons of general characteristics between the participants with gradable and ungradable photographs were presented in the Supplementary Table 3. Compared with those with gradable photos (n = 50,564, 96.99%), those with ungradable photos (n = 1570, 3.1%) were older, having longer diabetes duration, and worse control of glycemia (Supplementary Table 3). Thus, the estimated DR proportion in this group might be a bit higher. Finally, a total of 50,564 participants with gradable photographs were included in this study analysis.

## Data collection

Information on demographics, socio-economic status, lifestyle, family history of diseases, and medical history was collected. A metabolic equivalent was calculated according to moderate- and vigorous-intensity physical activity for work, in-transit, and leisure time throughout a whole week[29]. After an overnight fast of at least ten hours, blood samples were collected. Fasting plasma glucose was tested in local laboratories with unified quality control. HbA1c and serum lipids were centrally measured. The blood and urine specimens were stored and then shipped at a temperature range of 2–8 °C to the Guangzhou KingMed Diagnostics Group Co., Ltd. (Guangzhou, China)

for testing after the completion of the survey in one neighborhood or village[26]. BP, height, and weight were measured according to the standard protocol[26].

Presenting visual acuity proposed by the World Health Organization (WHO) was examined[30], with the logarithm of the minimal angle of resolution (log-MAR) charts used at a distance of five meters with each eye tested separately. Participants were seated in a windowless room with the lights turned off to allow the pupils to dilate naturally. Two 45-degree color fundus photographs were taken for each eye; one centered on the optic disc and the second on the macula, using a digital non-mydriatic retinal camera. The team grading the photos in this study consisted of eight ophthalmologists working in the ophthalmology department of the Shanghai Sixth People's Hospital affiliated to Shanghai Jiao Tong University School of Medicine. All of them received standardized training before the survey. Two qualified ophthalmologists graded each photograph, and a third ophthalmologist audited inconsistent results. Masking was adopted at each stage of evaluation.

The investigation period of 488 neighborhoods or rural villages of 122 study sites across 31 provinces were presented in Supplementary Table 4.

## Definition

The 31 provinces were divided into the northern or southern regions along the Qinling Mountains–Huaihe River Line[31]. Education levels were categorized into middle school and below as well as high school and above. Current smokers and drinkers were classified by whether they smoked or consumed alcohol during the questionnaire interview. Metabolic equivalents were calculated to express the intensity of physical activities based on the questionnaire collecting participants' activity types and time, including work, in-transit, and leisure time in a typical week. Moderate-intensity physical activity (MET value = 4.0) was defined as a

**Table 6 | Multivariable-adjusted odds ratios of demographic and clinical factors with any DR and VTDR**

| Independent variables | Any DR | | Any DR | | | |
|---|---|---|---|---|---|---|
| | OR (95% CI)[a] | $p$[a] | Non-VTDR | | VTDR | |
| | | | OR (95% CI)[a] | $p$[a] | OR (95% CI)[a] | $p$[a] |
| Demographics | | | | | | |
| Sex | | | | | | |
| Female vs male | 0.78 (0.69–0.87) | <0.0001 | 0.82 (0.72–0.92) | 0.0011 | 0.63 (0.53–0.75) | <0.0001 |
| Age, y | 1.00 (0.99–1.00) | 0.3435 | 1.00 (0.99–1.01) | 0.9652 | 0.98 (0.97–0.99) | 0.0008 |
| Geographical region | | | | | | |
| North vs south | 1.39 (1.22–1.58) | <0.0001 | 1.35 (1.17–1.55) | <0.0001 | 1.60 (1.34–1.91) | <0.0001 |
| Setting | | | | | | |
| Urban vs rural | 1.05 (0.88–1.26) | 0.5810 | 1.02 (0.86–1.22) | 0.8013 | 1.18 (0.91–1.53) | 0.2155 |
| Socio-economic status | | | | | | |
| Education levels | | | | | | |
| High school and above vs middle school or below | 0.82 (0.73–0.92) | 0.0008 | 0.84 (0.74–0.95) | 0.0041 | 0.77 (0.61–0.96) | 0.0211 |
| Average annual household income per capita, RMB | | | | | | |
| 10,000- < 20,000 vs <10,000 | 0.90 (0.79–1.03) | 0.1132 | 0.92 (0.80–1.05) | 0.2148 | 0.83 (0.64–1.07) | 0.1457 |
| ≥20,000 vs <10,000 | 0.76 (0.67–0.87) | <0.0001 | 0.79 (0.69–0.91) | 0.0008 | 0.65 (0.53–0.81) | 0.0001 |
| Unwilling to disclose vs <10,000 | 0.93 (0.82–1.06) | 0.3029 | 0.97 (0.85–1.12) | 0.7059 | 0.78 (0.66–0.92) | 0.0036 |
| Clinical characteristics | | | | | | |
| Diabetes duration, y | 1.08 (1.07–1.08) | <0.0001 | 1.06 (1.06–1.07) | <0.0001 | 1.12 (1.11–1.13) | <0.0001 |
| Family history of diabetes | | | | | | |
| Yes vs no | 1.13 (1.04–1.24) | 0.0050 | 1.12 (1.02–1.23) | 0.0190 | 1.21 (1.05–1.39) | 0.0063 |
| Attainment of targets | | | | | | |
| HbA1c, % | | | | | | |
| ≥7.0 vs <7.0 | 2.25 (2.03–2.50) | <0.0001 | 2.17 (1.94–2.43) | <0.0001 | 2.72 (2.27–3.27) | <0.0001 |
| BP, mmHg | | | | | | |
| ≥130/80 vs <130/80 | 1.30 (1.17–1.44) | <0.0001 | 1.29 (1.15–1.45) | <0.0001 | 1.33 (1.13–1.57) | 0.0004 |
| LDL-C, mmol/L | | | | | | |
| ≥2.6 vs <2.6 | 1.07 (0.97–1.17) | 0.1734 | 1.05 (0.95–1.15) | 0.3532 | 1.15 (0.96–1.37) | 0.1240 |

*Any DR* any diabetic retinopathy. *VTDR* vision-threatening diabetic retinopathy. *HbA1c* hemoglobin A1c. *BP* blood pressure. *LDL-C* low-density lipoprotein cholesterol. *MET* metabolic equivalent. *OR* odds ratio. *CI* confidence interval.

[a]The binary or multinomial surveylogistic regression (for two-sided test) was applied to calculate OR (95% CIs) and $p$ value, respectively when the dependent variables were the presence or absence of any DR or the severity of DR (absence of any DR, presence of non-VTDR, and presence of VTDR). Reference category was no DR. In the models, the independent variables were simultaneously included with the entered method as follows: the independent variables presented in Tables 6 and 7, body-mass index, smoking and drinking status, use of glucose-lowering treatment, antihypertensive treatment, lipid-lowering treatment, and attainment of high-density lipoprotein cholesterol and triglycerides.

Source data are provided as a Source Data file.

moderate amount of effort needed and noticeably accelerating the heart rate, while high-intensity physical activity (MET value = 8.0) was described as a large amount of effort required and causing rapid breathing and a substantial increase in heart rate during physical activities[29]. The classifications of physical activities were presented in detail in the protocol published before[26]. The adequate physical activity used in this study was defined as ≥600 MET minutes per week according to the Global Physical Activity Questionnaire analysis guide[29]. A family history of diabetes was identified if the participant answered that his or her first-degree relatives had diabetes. Metabolic control targets were defined as HbA1c < 7%, BP < 130/80 mmHg, and LDL-C < 2.6 mmol/L[32].

**Study-outcome definitions**

The primary outcomes were the presence and severity of any DR; the secondary outcome was distant vision impairment and blindness. The presence and severity of DR were identified and graded as no apparent retinopathy, mild, moderate, and severe non-proliferative diabetic retinopathy (NPDR), and PDR[33]. DME was considered to be present when there was retinal thickening at or within one disc diameter of the macular center or definite hard exudates in this region[34]. Furthermore, clinically significant macular edema (CSME) was

identified if any of the following characteristics was present: retinal thickening at or within 500 μm of the macular center; hard exudates at or within 500 μm of the macular center with adjacent retinal thickening; or retinal thickening of one disk area or greater in size, at least part of which was within one disc diameter of the macular center[34]. Gradings were defined according to the most severe grade of the fundus photographs in both eyes of each patient. Any DR was defined as presence of non-proliferative DR, proliferative DR, diabetic maculopathy, or a combination thereof[35]. Then, any DR was divided into two categories: non-VTDR and VTDR. Non-VTDR included mild and moderate non-proliferative DR, DME except CSME, or a combination thereof. VTDR was defined as presence of severe NPDR, PDR, CSME, or a combination thereof[35]. The distant visual impairment was categorized based on the WHO standards of blindness and vision impairment[36].

**Statistical analysis**

Descriptive data were presented as median (25th percentile-75th percentile) for continuous variables and number (proportion) for categorical variables. Statistical analyses were performed considering strata, cluster, and weight variables to accommodate the sampling scheme unless stated otherwise. The sex-, age-, and urban/

**Table 7 | Multivariable-adjusted odds ratios of lifestyle factors with any DR and VTDR**

| Independent variables | Any DR | | Any DR | | | |
|---|---|---|---|---|---|---|
| | OR (95% CI)[a] | p[a] | Non-VTDR | | VTDR | |
| | | | OR (95% CI)[a] | p[a] | OR (95% CI)[a] | p[a] |
| Physical activity, MET min/week | | | | | | |
| ≥600 vs <600 | 0.88 (0.77–0.99) | 0.0375 | 0.88 (0.76–1.02) | 0.0896 | 0.87 (0.73–1.03) | 0.1023 |
| Dietary intake[b] | | | | | | |
| Refined grains, g/day | | | | | | |
| >450 vs ≤450 | 1.06 (0.97–1.16) | 0.2040 | 1.11 (1.01–1.22) | 0.0213 | 0.85 (0.69–1.06) | 0.1405 |
| Whole grains, g/day | | | | | | |
| >50 vs ≤50 | 1.03 (0.94–1.13) | 0.4674 | 1.02 (0.93–1.13) | 0.6167 | 1.08 (0.91–1.28) | 0.3602 |
| Potatoes, g/day | | | | | | |
| >31 vs ≤31 | 0.86 (0.79–0.94) | 0.0003 | 0.84 (0.78–0.91) | <0.0001 | 0.94 (0.77–1.14) | 0.5201 |
| Soybean products, g/day | | | | | | |
| >30 vs ≤30 | 1.00 (0.89–1.12) | 0.9837 | 1.00 (0.90–1.12) | 0.9440 | 0.99 (0.79–1.26) | 0.9557 |
| Fresh vegetables, g/day | | | | | | |
| >500 vs ≤500 | 1.12 (1.00–1.25) | 0.0570 | 1.11 (0.98–1.26) | 0.0966 | 1.14 (0.95–1.37) | 0.1631 |
| Fresh fruits, g/day | | | | | | |
| >100 vs ≤100 | 0.85 (0.77–0.95) | 0.0039 | 0.87 (0.77–0.99) | 0.0255 | 0.77 (0.63–0.95) | 0.0113 |
| Dairy products, ml/day | | | | | | |
| >100 vs ≤100 | 0.93 (0.86–1.00) | 0.0594 | 0.90 (0.84–0.98) | 0.0121 | 1.03 (0.85–1.26) | 0.7447 |
| Red meat, g/day | | | | | | |
| >100 vs ≤100 | 1.09 (0.97–1.21) | 0.1345 | 1.10 (0.98–1.22) | 0.0950 | 1.04 (0.85–1.27) | 0.7089 |
| Poultry, g/day | | | | | | |
| >15 vs ≤15 | 0.95 (0.86–1.05) | 0.2885 | 0.98 (0.88–1.09) | 0.6729 | 0.82 (0.64–1.05) | 0.1133 |
| Seafood, g/day | | | | | | |
| >26 vs ≤26 | 0.95 (0.85–1.06) | 0.3175 | 0.96 (0.85–1.08) | 0.4893 | 0.89 (0.75–1.05) | 0.1679 |
| Eggs, g/day | | | | | | |
| >50 vs ≤50 | 0.98 (0.89–1.07) | 0.6114 | 0.96 (0.88–1.05) | 0.3374 | 1.06 (0.85–1.30) | 0.6100 |
| Nuts | | | | | | |
| Yes vs no | 0.94 (0.85–1.05) | 0.2805 | 0.96 (0.86–1.07) | 0.4297 | 0.89 (0.75–1.05) | 0.1522 |
| Fresh juices | | | | | | |
| Yes vs no | 0.93 (0.76–1.12) | 0.4158 | 0.89 (0.73–1.08) | 0.2365 | 1.09 (0.73–1.62) | 0.6614 |

*Any DR* any diabetic retinopathy. *VTDR* vision-threatening diabetic retinopathy. *HbA1c* hemoglobin A1c. *BP* blood pressure. *LDL*-C low-density lipoprotein cholesterol. *MET* metabolic equivalent. *OR* odds ratio. *CI* confidence interval.

[a]The binary or multinomial surveylogistic regression (for two-sided test) was applied to calculate OR (95% CIs) and p value, respectively when the dependent variables were the presence or absence of any DR or the severity of DR (absence of any DR, presence of non-VTDR, and presence of VTDR). Reference category was no DR. In the models, the independent variables were simultaneously included with the entered method as follows: the independent variables presented in Tables 6 and 7, body-mass index, smoking and drinking status, use of glucose-lowering treatment, antihypertensive treatment, lipid-lowering treatment, and attainment of high-density lipoprotein cholesterol and triglycerides.

[b]The consumption values of 13 food groups were categorical variables. The integer quartile cutoffs of consumption of the 11 food groups (refined grains, whole grains, potatoes, soybean products, fresh vegetables, fresh fruits, dairy products, red meat, poultry, seafood, and eggs) were selected to define high or low consumption. For nuts and fresh juices, consumption or non-consumption was defined.

Source data are provided as a Source Data file.

rural structure of adults with diabetes aged 18–74 years in China in 2018–2019 from the CCDRFS dataset (Supplementary Table 5) was used as a reference population for weighting frequency, and also was used to estimate 2018–2019 diabetes prevalence[7]. The standardized province-specific prevalence was calculated according to the sex- and age- structure of the reference using the direct standardized method.

Differences in medians or proportions between the two groups were tested using the Wilcoxon rank test or the chi-square test. The linear trend of proportions was analyzed using a logistic regression model with the median value of each subgroup representing the group level. The odds ratio (OR) and its 95% confidence interval (CI) of DR presence or severity with related factors were evaluated using the multivariable binary and multinomial logistic models, respectively. Cases with complete data on primary outcomes were used for analysis due to the small number of missing values.

Data analyses were conducted using SAS (version 9.4, SAS Institute). All tests were two-sided, and a $p < 0.050$ was considered statistically significant.

## Data availability

The export of human-related data is governed by the Ministry of Science and Technology of China (MOST) and must adhere to the Regulations of the People's Republic of China on Administration of Human Genetic Resources (State Council No.717). Request for the non-profit use of the dataset of the China National Diabetic Chronic Complications Study should be sent to the corresponding author Weiping Jia. The requests for the data will be replied to within 10 business days. Furthermore, the joint application for the data sharing by the corresponding author combined with the data requester should then be submitted to MOST. Upon approval from MOST, the data can be provided to the requester. The relevant data are available within the

Article, Supplementary Information, or Source Data file. Source data are provided with this paper.

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

## Acknowledgements

We thank all investigators and all participants for their contributions to the China National Diabetic Chronic Complications Study. We are grateful for the support from the Bethune Charitable Foundation. This work was also supported by grants from the Shanghai Science and Technology Committee (grant No. 19692115900 and 17411952600), Shanghai Municipal Key Clinical Specialty, and the Chinese Academy of Engineering (grant No. 2022-XY-08) to W.J., the National Key Research and Development Program of China (grant No. 2021YFC2500201) to L.W., and the Strategic Priority Research Program of the Chinese Academy of Sciences (grant No. XDB38020000) to J.W (Jiarui Wu).

## Author contributions

W.J. conceived and supervised the study and provided scientific direction. D.Z. (Dalong Zhu), L.G., J.W. (Jianping Weng), Z.Z., D.Z. (Dajin Zou), Q.J., X.G., Q.W., Z.H. collected the data. M.Z., S.C., R.Y., H.C., and X.Z. performed the statistical analysis. X.H., and L.W. interpreted the results and drafted the manuscript. J.W. (Jiarui Wu) and J.W. (Jing Wu) provided critical comments and reviewed the manuscript. All authors revised the manuscript and approved the final version before submission.

## Competing interests

The authors declare no competing interests.

## Additional information

[1]Department of Endocrinology and Metabolism, Shanghai Sixth People's Hospital Affiliated to Shanghai Jiao Tong University School of Medicine, Shanghai Diabetes Institute, Shanghai Key Laboratory of Diabetes Mellitus, Shanghai Clinical Center for Diabetes, Shanghai Key Clinical Center for Metabolic Disease, Shanghai, China. [2]National Center for Chronic and Noncommunicable Disease Control and Prevention, Chinese Center for Disease Control and Prevention, Beijing, China. [3]Department of Endocrinology, Drum Tower Hospital Affiliated to Nanjing University Medical School, Nanjing, Jiangsu Province, China. [4]Department of Endocrinology, Beijing Hospital, Beijing, China. [5]Department of Endocrinology, the First Affiliated Hospital, Division of Life Sciences and Medicine, University of Science and Technology of China, Hefei, Anhui Province, China. [6]Institute of Metabolism and Endocrinology, Key Laboratory of Diabetes Immunology, Ministry of Education, National Clinical Research Center for Metabolic Diseases, the Second Xiangya Hospital and the Diabetes Center, Central South University, Changsha, Hunan Province, China. [7]Department of Endocrinology, Changhai Hospital, Second Military Medical University, Shanghai, China. [8]Department of Endocrinology, Xijing Hospital, Xi'an, Shaanxi Province, China. [9]Department of Endocrinology, Peking University First Hospital, Beijing, China. [10]Department of Ophthalmology, Shanghai Jiao Tong University Affiliated Sixth People's Hospital, Shanghai, China. [11]Department of Endocrinology and Metabolism, Suzhou Dushu Lake Hospital (Dushu Lake Hospital Affiliated to Soochow University), Suzhou, Jiangsu Province, China. [12]Key Laboratory of Systems Health Science of Zhejiang Province, Hangzhou Institute for Advanced Study, University of Chinese Academy of Sciences, Hangzhou, Zhejiang Province, China. [13]Center for Excellence in Molecular Science, Chinese Academy of Sciences, Shanghai, China. [14]The authors contributed equally: Xuhong Hou, Limin Wang. *A list of authors and their affiliations appears at the end of the paper. ✉e-mail: wujing@chinacdc.cn; wpjia@sjtu.edu.cn

## for the China National Diabetic Chronic Complications (DiaChronic) Study Group

**Xuhong Hou[1,14], Weiping Jia ®[1]✉, Dalong Zhu[3], Jianping Weng ®[5], Zhiguang Zhou[6] & Xiaohui Guo[9]**

A full list of members and their affiliations appears in the Supplementary Information.

