## [Peer Review File · Nature Communications]

Response to Comments from the Reviewers:

Reviewer #1:

This is a nation-wide assessment of the prevalence of diabetic retinopathy and its various stages among individuals with diabetes in China. The main question is about the novelty of its valid findings.

Author response: Thanks for your comments. We agree that novelty is essential for a research. To the best of our knowledge, our study is the first nation-wide prevalence survey, systematically investigating the distribution of diabetic retinopathy (DR) and vision-threatening diabetic retinopathy (VTDR), and associated risk factors, across all thirty-one provinces in China. Strengths of our study compared with previous studies are listed as below.

Firstly, as we know, so far only two multiple province-level prevalence surveys of DR were conducted between 2014 and 2018^{1,2}. 1) One involved 6 provinces and the other involved 12 provinces. However, our study covered all the 31 provinces nationwide. 2) The previous two studies recruited participants from both hospitals and communities, which may introduce bias of DR-data in this study, resulting in a lack of representativeness. On the other hand, our data was all collected from communities to avoid such bias. 3) In this study, we collaborated with the Chinese Center for Disease Control and Prevention (China CDC). All the staff in this survey were professional, and familiar with local situation. Thus, all these guarantee the quality of data collected and representativeness in this survey.

Secondly, our study consisted of a systematic and comprehensive investigation of the associated risk factors of DR, including not only information on socio-demographics, medical history, and clinical data, but also information of lifestyle factors, including physical activity indicated by the calculation of metabolic equivalent and detailed dietary intake categories. So far, no report of the associations of lifestyle factors with DR nationwide from China has ever been published before. Based on the detailed data of associated factors collected in this study, we reported several important findings.

Taken together, our study is the biggest survey collecting qualified data from all the provinces nationwide, collecting comprehensive data and reporting the latest DR prevalence, distributions and associated factors from China. In order to enhance these points, we have revised the following parts accordingly and listed the revised parts as follows.

“Introduction” Line 85-89: “...As we know, so far only two multiple province-level prevalence surveys of DR were conducted in China between 2014 and 2018, with one involving 6 provinces and the other involving 12 provinces ^{8,9}. These two studies recruited participants from both hospitals and communities, which might introduce selection bias in the studies, resulting in a lack of representativeness.”

“Discussion” Line 290-297: “...Firstly, it was the first nation-wide, population-based survey of DR, with a multistage sampling scheme, including stratification, clustering, and randomization, through the disease surveillance system (including urban and rural sites) from the China Chronic Disease and Risk Factors Surveillance. Together with a systematic and comprehensive investigation of the associated risk factors, including not only information on socio-demographics, medical history, and clinical data, but also detailed information of lifestyle, it makes it possible to describe the patterns of influencing factors associated with different DR prevalence....”

1. Line 54: The national prevalence of diabetic retinopathy (DR) and vision-threatening DR (VTDR) was 16.3% (95% confidence interval [CI] 15.3%- 56 17.2%) and 3.2% (2.9%-3.5%), respectively. It may be added that the figures refer to individuals with diabetes.

Author response: Thanks for your comments. We have revised as suggested and listed it as below.

“Abstract” Line 53-56: “The prevalence of diabetic retinopathy (DR) and vision-threatening DR (VTDR) among individuals with diabetes were 16.3% (95% confidence interval [CI] 15.3%-17.2%) and 3.2% (2.9%-3.5%)...”

2. Line 59: The differences in prevalence of any DR and VTDR between those who have attained a given metabolic goal and those who have not were more pronounced for Hemoglobin A1c than for blood pressure and low density lipoprotein cholesterol. Hard data should be presented.

Author response: Thanks for your comments. We have added these data accordingly and listed it as below.

“Abstract” Line 57-61: “The significant differences in prevalence between those who have not attained a given metabolic goal and those who have were more

pronounced for Hemoglobin A1c (any DR: 22.3% vs 8.7%, VTDR: 4.6% vs 1.4%) than for blood pressure (any DR: 17.5% vs 13.7%, VTDR: 3.4% vs 2.7%) and low-density lipoprotein cholesterol (any DR: 16.7% vs 15.4%).”

3. *Line 62: Multiple factors were associated with the presence and severity of DR. Which factors, and how were the statistical data?*

Author response: Thanks for your comments. We have added detailed description of multiple factors in the result part, listed as below. Due to the limited number of abstract (no more than 150 words) required by the journal format, we could only make a brief summary of the multivariable analysis in the abstract as before.

“Results” Line 165-197: “Multivariable analyses results assessing the factors associated with any DR and the severity of DR (non-VTDR and VTDR) were shown in Table 5. In terms of demographic and socio-economic factors, after adjusting for confounding factors, it showed that any DR, non-VTDR and VTDR were all negatively associated with females, with ORs ranging from 0.63 (95% CI 0.53-0.75) to 0.82 (95% CI 0.72-0.92), education levels of high school and above, with ORs ranging from 0.77 (95% CI 0.61-0.96) to 0.84 (95% CI 0.74-0.95), and average annual household income per capita over ¥20,000, with ORs ranging from 0.65 (95% CI 0.53-0.81) to 0.79 (95% CI 0.69-0.91). Age was only significantly negatively associated with VTDR (OR 0.98, 95% CI 0.97-0.99). In terms of geographical region, it seemed that people living in the northern region in China were more likely to have any DR (OR 1.39, 95% CI 1.22-1.58), non-VTDR (OR 1.35, 95% CI 1.17-1.55) and VTDR (OR 1.60, 95% CI 1.34-1.91).

In terms of clinical characteristics, through multivariable analyses, it showed that diabetes duration and family histories were both positively associated with any DR, non-VTDR and VTDR, with ORs ranging from 1.06 (95% CI 1.06-1.07) to 1.12 (95% CI 1.11-1.13) and from 1.12 (95% CI 1.02-1.23) and 1.21 (95% CI 1.05-1.39), respectively. In addition, not achieving the targets for HbA1c and BP were both positively associated with any DR, non-VTDR, and VTDR, with ORs ranging from 2.17 (95% CI 1.94-2.43) to 2.72 (95% CI 2.27-3.27) and from 1.29 (95% CI 1.15-1.45) to 1.33 (95% CI 1.13-1.57), respectively. Nevertheless, there was no significant association between not achieving the target for LDL-C and any DR, non-VTDR, and VTDR.

Further, detailed analyses on the association of diet and physical activity with any DR, non-VTDR, and VTDR were completed in this nationwide study. It found that

physical activity over 600 MET minutes/week were significantly negatively associated with any DR (OR 0.88, 95% CI 0.77-0.99) only. In terms of diet, fresh fruits > 100 g/day were protective factors for any DR (OR 0.85, 95% CI 0.77-0.95), non-VTDR (OR 0.87, 95% CI 0.77-0.99) and VTDR (OR 0.77, 95% CI 0.63-0.95). Potatoes > 31 g/day were protective factors for any DR and non-VTDR, with ORs of 0.86 (95% CI 0.79-0.94) and 0.84 (95% CI 0.78-0.91), while dairy products > 100 ml/day was only positively associated with non-VTDR (OR 0.90, 95% CI 0.84-0.98). Nevertheless, refined grains >450 g/day was only positively associated with non-VTDR (OR 1.11, 95% CI 1.01-1.22).”

4. The Abstract should report which novel findings were obtained in the study

Author response: Thanks for your comments. We have listed the main novel findings as follows, including the first nation-wide prevalence of DR ever completed in China, and the findings of significant differences of prevalence between the northern and southern regions, and those who have not attained a given metabolic goal and those who have. We revised the information as suggested above in the abstract and listed as below.

“Abstract” Line 51-62: “The first national survey of diabetic complications was conducted in Chinese adults with diabetes between 2018 and 2020. Through a multistage sampling scheme, 50564 participants with gradable non-mydratiac fundus photographs were analyzed. The prevalence of diabetic retinopathy (DR) and vision-threatening DR (VTDR) among individuals with diabetes were 16.3% (95% confidence interval [CI] 15.3%-17.2%) and 3.2% (2.9%-3.5%), significantly higher in the northern (DR 18.1%; VTDR 3.8%) than in the southern (DR 14.4%; VTDR 2.5%) regions. The significant differences in prevalence between those who have not attained a given metabolic goal and those who have were more pronounced for Hemoglobin A1c (any DR: 22.3% vs 8.7%, VTDR: 4.6% vs 1.4%) than for blood pressure (any DR: 17.5% vs 13.7%, VTDR: 3.4% vs 2.7%) and low-density lipoprotein cholesterol (any DR: 16.7% vs 15.4%). Multiple factors were associated with the presence and the severity of DR.”

5. Line 68: DR is often asymptomatic and brings irreversible vision impairment. The statements are contradictory.

Author Response: Thanks for your comments. DR is often insidious and asymptomatic at early stages, usually Stages 1-2. It usually results in the unawareness of DR during the early stages for people with diabetes. This might lead to irreversible vision impairment, quickly progressing into VTDR, Stages 3-5, without early finding and treatment. This also highlights the importance of promoting DR screening among people with diabetes. We have revised the sentence accordingly to avoid misunderstanding and listed it as below.

“Introduction” Line 68-71: “Although DR is often insidious and asymptomatic at early stages, it might quickly progress into VTDR without awareness and intervention on metabolic risk factors, and then could lead to irreversible vision impairment.”

6. *Line 81: However, the relevant studies are badly lacking in Asia, China is the country worldwide with the highest number of recent epidemiological studies in the field of ophthalmology.*

Author Response: Thanks for your comments. We have revised the sentence and listed it as below.

“Introduction” Line 82-84: “With the largest number of people with diabetes, around one-fourth of the global number, living in China, there is a lack of the latest data representing nation-wide status of DR to guide the prevention and control strategy ⁷.”

7. *Line 289: The sentence The study protocol and data collection were presented in detail in the paper and discussed briefly below. may be re-worded.*

Author response: Thanks for your comments. We have revised the sentence and listed it as below.

“Methods” Line 321: “The study protocol has been published before ²⁶ and summarized briefly below.”

8. *Line 303: The comparisons of general characteristics between the participants with gradable and ungradable photographs were presented in Supplementary Table 3. The*

busy reader would like to read the information directly in the main text and not to have to turn to the supplementary material.

Author response: Thanks for your comments. We have added a brief description of Supplementary Table 3 in the manuscript as follows.

“Results” Line 110-114: “Compared with those with gradable photos (n=50564, 96.99%), those with ungradable photos (n=1570, 3.01%) were older, having longer diabetes duration, and worse control of glycemia (Supplementary Table 3). Thus, the estimated DR proportion in this group might be a bit higher. However, due to the very low proportion of ungradable photos (3.01%) among all data, the effect was minimal.”

9. Line 292: It has remained unclear for this reviewer, (1) how diabetes mellitus was diagnosed and (2) how the 58560 participants were sampled from the diabetes management registration system of 488 neighborhoods/villages? (3) And how were the neighborhoods selected?

Author response: Thanks for your comments. The responses to the three questions were as follows.

(1) how diabetes mellitus was diagnosed

All those participants recruited in this study were people with diabetes diagnosed by physicians in hospitals, registered in the diabetes management registration system of basic public health services ³ in community health centers and monitored by the local Center for Disease Control and Prevention (CDC). We have added the description of this in the manuscript as follows.

“Methods” Line 326-330: “All those recruited in this study were people with diabetes diagnosed by physicians in hospitals, registered in the diabetes management registration system of basic public health services ²⁷ in community health centers and monitored by the local Center for Disease Control and Prevention.”

(2) how the 58560 participants were sampled from the diabetes management registration system of 488 neighborhoods/villages? (3) And how were the neighborhoods selected?

A flowchart of how the 58560 participants were sampled through a multistage sampling scheme (stratification, clustering, and random) was published before in the protocol of this study ⁴. Briefly, there are three stages of stratification. In the first stage, generally four study sites based on the disease surveillance points were selected from each province. Finally, a total of 122 study sites (65 urban study sites and 57 rural study sites) were randomly selected and invited to participate. In the second stage, four neighborhoods in urban areas or four villages in rural areas were randomly selected from each study site, resulting in 260 neighborhoods and 228 villages in total. In the third stage, according to the age and gender distribution by the CCDRFS 2013 diabetes data, the national sample size of 58,560 individuals and the sample size of 480 at each study site were set. Finally, 480 participants were randomly invited from those registered in the diabetes management registration system among the four neighborhoods and/or villages at each study site. About 1.68 million patients with diagnosed diabetes aged 18-75 years were registered in the diabetes management registration systems at the 122 study sites during survey sampling. We have added a brief introduction of this in the manuscript and listed it as follows.

“Methods” Line 335-345: “A flowchart of the multistage sampling scheme was listed in the protocol of this study ²⁶. Briefly speaking, there are three stages of stratification. In the first stage, four study sites based on the disease surveillance points were selected from each province generally. Finally, a total of 122 study sites (65 urban study sites and 57 rural study sites) were randomly selected and invited to participate. In the second stage, four neighborhoods in urban areas or four villages in rural areas were randomly selected from each study site, resulting in 260 neighborhoods and 228 villages in total. In the third stage, according to the age and gender distribution by the CCDRFS 2013 diabetes data, the national sample size of 58,560 individuals and the sample size of 480 at each study site were set. 480 participants were randomly invited from those registered in the diabetes management registration system at each study site.”

10. Line 316: Hemoglobin A1c (HbA1c) and serum lipids were centrally measured. Were the blood samples mailed to a central laboratory?

Author response: Thanks for your comments. After the completion of the survey in one neighborhood or village, the blood and urine specimens were stored and shipped at a temperature range of 2-8 °C to the Guangzhou KingMed Diagnostics Group Co., Ltd.

(Guangzhou, China) for testing as described in the protocol of this study ⁴. We have added a brief introduction of this in the manuscript and listed it as follows.

“Methods” Line 362-364: “After the completion of the survey in one neighborhood or village, the blood and urine specimens were stored and shipped at a temperature range of 2-8 °C to the Guangzhou KingMed Diagnostics Group Co., Ltd. (Guangzhou, China) for testing ²⁶.”

11. Line 319: Distant visual acuity was examined. How, uncorrected, pinhole or corrected?

Author response: Thanks for your comments. Presenting visual acuity (PVA) proposed by WHO was examined in this study ^{5,6}. PVA is defined as the uncorrected visual acuity of those who do not wear corrective spectacles, or the corrected visual acuity of those who wear spectacles in their daily life. We have revised related description and listed it as follows.

“Methods” Line 367-369: “Presenting visual acuity proposed by WHO was examined ^{30,31}, with the logarithm of the minimal angle of resolution (log-MAR) charts used at a distance of five meters with each eye tested separately.”

12. 324: The word Blinding may be replaced by Masking.

Author response: Thanks for your comments. We have replaced blinding with masking in the manuscript accordingly and listed it as follows.

“Methods” Line 377-378: “Masking was adopted at each stage of evaluation.”

13. Line 330: A metabolic equivalent was calculated throughout a week and adequate physical activity was defined according to the Global Physical Activity Questionnaire analysis guide The metabolic equivalent should be explained in greater detail.

Author response: Thanks for your comments. A metabolic equivalent was calculated to express the intensity of physical activities based on the questionnaire collecting participants’ activity kinds and time including work, in-transit, and leisure time in a typical week. Moderate-intensity physical activity (MET value = 4.0) was defined as a moderate amount of effort needed and noticeably accelerating the heart rate, while high-intensity physical activity (MET value=8.0) was defined as a large amount of effort

needed and causing rapid breathing and a substantial increase in heart rate ⁷. The classifications of physical activities were presented in detail in the protocol published before ⁴. The adequate physical activity used in this study was defined as ≥ 600 MET minutes per week according to the Global Physical Activity Questionnaire analysis guide ⁷. We have revised related description in the manuscript and listed it as follows.

“Methods” Line 384-393: “A metabolic equivalent was calculated to express the intensity of physical activities based on the questionnaire collecting participants’ activity kinds and time including work, in-transit, and leisure time in a typical week. Moderate-intensity physical activity (MET value = 4.0) was defined as a moderate amount of effort needed and noticeably accelerating the heart rate, while high-intensity physical activity (MET value=8.0) was defined as a large amount of effort needed and causing rapid breathing and a substantial increase in heart rate during physical activities ²⁹. The classifications of physical activities were presented in detail in the protocol published before ²⁶. The adequate physical activity used in this study was defined as ≥ 600 MET minutes per week according to the Global Physical Activity Questionnaire analysis guide ²⁹. ”

14. Line 109: It should be pointed out that these were the results of a univariate analysis?

Author response: Thanks for your comments. We have revised it accordingly and listed it as follows.

“Results” Line 115-117: “Univariate analyses showed that compared with the participants without DR, those with any DR had significantly higher proportions of Northerners and longer diabetes duration, but lower education and income levels.”

15. Line 154: The rates of unilateral and bilateral blindness among the patients with VTDR were 8.23-fold (95% CI 6.01-11.26) and 9.72- fold (5.17-18.28) higher than those with non-VTDR, respectively, after adjustment for sex and age (Table 4). Is there information on the other causes of blindness?

Author response 1: Thanks for your comments. Through the check of questionnaire and ophthalmologists' records, the other cause-related blindness recorded in this study included cataracts, eye trauma, high myopia, keratopathy (keratitis, corneal degeneration, and corneal dystrophy), retinopathy (macular degeneration, retinal detachment), optic neuropathy, choroidopathy, glaucoma, strabismus, vitreous diseases

(vitreous opacity, vitreous hemorrhage), nystagmus, presbyopia, ocular tumors, pterygium, amblyopia, intraocular lens dislocation, congenital and hereditary eye diseases, measles sequela, ocular tumors, and other diseases (cerebral infarction, sequela of cerebral infarction). Thus, we recompleted the analyses of table 4 by enrolling 67 additional participants who self-reported blindness with unknown causes (without completing distant acuity examination) and having fundus photography taken first, and then excluding 729 participants for the analysis of worse-seeing eye and 93 participants for the analysis of better-seeing eye, respectively, due to one or more of the aforementioned other cause-related blindness. The revised table 4 was listed as follows, and we have revised the description in the manuscript accordingly.

“Results” Line 160-163: “The rates of worse-seeing and better-seeing eye blindness among the patients with VTDR were 11.25-fold (95% CI 8.13-15.58) and 10.26-fold (95% CI 5.97-17.65) higher than those with non-VTDR, respectively, after adjustment for sex and age (Table 4).”

“Discussion” Line 246-249: “In line with these results, our study observed that the proportions of worse-seeing and better-seeing eye blindness were 11.25-fold (95% CI 8.13-15.58) and 10.26-fold (95% CI 5.97-17.65) higher for patients with VTDR than those with non-VTDR.”

Revised Table 4 Sex- and age-adjusted odds ratios of distant visual impairment associated with DR and VTDR.*

	Total	Any DR				Any DR			
		No DR	Any DR	OR (95% CI) [†]	p [†]	Non-VTDR	VTDR	OR (95% CI) [†]	p [†]
Worse-seeing eye [‡]									
N	49600	41261	8339			6744	1595		
Normal	34058 (68.7)	29085 (70.5)	4973 (59.6)	1 (ref)		4367 (64.8)	606 (38.0)	1 (ref)	
Mild	4641 (9.4)	3748 (9.1)	893 (10.7)	1.56 (1.33-1.84)	<0.0001	737 (10.9)	156 (9.8)	1.52 (1.08-2.14)	0.016
Moderate	9472 (19.1)	7459 (18.1)	2013 (24.1)	1.64 (1.46-1.83)	<0.0001	1420 (21.1)	593 (37.2)	2.83 (2.36-3.38)	<0.0001
Severe	453 (0.91)	325 (0.79)	128 (1.5)	3.09 (1.68-5.70)	0.00025	79 (1.2)	49 (3.1)	2.75 (1.13-6.72)	0.024
Blindness	976 (2.0)	644 (1.6)	332 (4.0)	3.08 (2.37-4.00)	<0.0001	141 (2.1)	191 (12.0)	11.25 (8.13-15.58)	<0.0001
Better-seeing eye [§]									
N	50236	41739	8497			6838	1659		
Normal	41747 (83.1)	35241 (84.4)	6506 (76.6)	1 (ref)		5510 (80.6)	996 (60.0)	1 (ref)	
Mild	3000 (6.0)	2374 (5.7)	626 (7.4)	1.58 (1.27-1.97)	<0.0001	485 (7.1)	141 (8.5)	1.42 (1.02-1.97)	0.034
Moderate	5073 (10.1)	3846 (9.2)	1227 (14.4)	1.84 (1.59-2.12)	<0.0001	786 (11.5)	441 (26.6)	2.73 (2.25-3.32)	<0.0001
Severe	150 (0.30)	109 (0.26)	41 (0.48)	4.60 (1.50-14.09)	0.0067	19 (0.28)	22 (1.3)	2.02 (0.53-7.78)	0.30
Blindness	266 (0.53)	169 (0.40)	97 (1.1)	3.07 (1.96-4.83)	<0.0001	38 (0.56)	59 (3.6)	10.26 (5.97-17.65)	<0.0001

DR=diabetic retinopathy. VTDR=vision-threatening diabetic retinopathy. OR=odds ratio. CI=confidence interval.

* Data were presented as number (percentage) unless otherwise stated. Percentages might not add to 100% because of rounding.

[†] OR (95% CI) and p were calculated by using a multinomial multivariable logistic regression with the severity of the distant visual impairment (normal, mild, moderate, severe, and blindness; normal as the referent category) as a dependent variable and the presence or absence of any DR or VTDR as the independent variable (presence versus absence), respectively, after adjusting for age and gender.

[‡] Data were analyzed after excluding participants with other cause-related blindness (N=729), including cataracts, eye trauma, high myopia, keratopathy (keratitis, corneal degeneration, and corneal dystrophy), retinopathy (macular degeneration, retinal detachment), optic neuropathy, choroidopathy, glaucoma, strabismus, vitreous diseases (vitreous opacity, vitreous hemorrhage), nystagmus, presbyopia, ocular tumors, pterygium, amblyopia, intraocular lens dislocation, congenital and hereditary eye diseases, measles sequela, and other diseases (cerebral infarction, sequela of cerebral infarction).

[§] Data were analyzed after excluding participants with other cause-related blindness (N=93), including high myopia, retinopathy (macular degeneration, retinal detachment), cataract, eye trauma, congenital and hereditary eye diseases, glaucoma, presbyopia, keratopathy (corneal degeneration), nystagmus, ocular tumors, vitreous diseases (vitreous opacity), choroidopathy, and other diseases (sequela of cerebral infarction).

16. Line 158: *The multivariable analysis should be described in greater detail.*

Author response: Thanks for your comments. We have revised the descriptions of the multivariable analysis, which are listed as below.

“Results” Line 165-197: “Multivariable analyses results assessing the factors associated with any DR and the severity of DR (non-VTDR and VTDR) were shown in Table 5. In terms of demographic and socio-economic factors, after adjusting for confounding factors, it showed that any DR, non-VTDR and VTDR were all negatively associated with females, with ORs ranging from 0.63 (95% CI 0.53-0.75) to 0.82 (95% CI 0.72-0.92), education levels of high school and above, with ORs ranging from 0.77 (95% CI 0.61-0.96) to 0.84 (95% CI 0.74-0.95), and average annual household income per capita over ¥20,000, with ORs ranging from 0.65 (95% CI 0.53-0.81) to 0.79 (95% CI 0.69-0.91). Age was only significantly negatively associated with VTDR (OR 0.98, 95% CI 0.97-0.99). In terms of geographical region, it seemed that people living in the northern region in China were more likely to have any DR (OR 1.39, 95% CI 1.22-1.58), non-VTDR (OR 1.35, 95% CI 1.17-1.55) and VTDR (OR 1.60, 95% CI 1.34-1.91).

In terms of clinical characteristics, through multivariable analyses, it showed that diabetes duration and family histories were both positively associated with any DR, non-VTDR and VTDR, with ORs ranging from 1.06 (95% CI 1.06-1.07) to 1.12 (95% CI 1.11-1.13) and from 1.12 (95% CI 1.02-1.23) and 1.21 (95% CI 1.05-1.39), respectively. In addition, not achieving the targets for HbA1c and BP were both positively associated with any DR, non-VTDR, and VTDR, with ORs ranging from 2.17 (95% CI 1.94-2.43) to 2.72 (95% CI 2.27-3.27) and from 1.29 (95% CI 1.15-1.45) to 1.33 (95% CI 1.13-1.57), respectively. Nevertheless, there was no significant association between not achieving the target for LDL-C and any DR, non-VTDR, and VTDR.

Further, detailed analyses on the association of diet and physical activity with any DR, non-VTDR, and VTDR were completed in this nationwide study. It found that physical activity over 600 MET minutes/week were significantly negatively associated with any DR (OR 0.88, 95% CI 0.77-0.99) only. In terms of diet, fresh fruits > 100 g/day were protective factors for any DR (OR 0.85, 95% CI 0.77-0.95), non-VTDR (OR 0.87, 95% CI 0.77-0.99) and VTDR (OR 0.77, 95% CI 0.63-0.95). Potatoes > 31 g/day were protective factors for any DR and non-VTDR, with ORs of 0.86 (95% CI 0.79-0.94) and 0.84 (95% CI 0.78-0.91), while dairy products > 100 ml/day was only positively associated with non-VTDR (OR 0.90, 95% CI 0.84-0.98).

Nevertheless, refined grains >450 g/day was only positively associated with non-VTDR (OR 1.11, 95% CI 1.01-1.22).”

17. *It appears that hyperopia or short axial length was not included into the analysis, although is it a major ocular risk parameter for DR?*

Author response: Thanks for your comments. This study is conducted in community health centers, mainly completed by community health staff and CDC staff, rather than being completed in ophthalmology specialized departments from hospitals. Although hyperopia or short axial length is one of the ocular risk parameters for DR, it is difficult to complete the assessment in a large-scale epidemiology study in community health centers. We have added this in the limitation part and revised it as follows.

“Discussion” Line 301-305: “Besides, this study is conducted in community health centers instead of being completed in ophthalmology specialized departments in hospitals. Due to the limited resources, it is difficult to include the assessment of some ocular risk parameters for DR, like hyperopia or short axial length, in a large-scale epidemiology study.”

References:

- 1 Li, M. *et al.* Females with Type 2 Diabetes Mellitus Are Prone to Diabetic Retinopathy: A Twelve-Province Cross-Sectional Study in China. *J Diabetes Res* **2020**, 5814296 (2020). <https://doi.org:10.1155/2020/5814296>
- 2 Liu, Y. *et al.* Prevalence of diabetic retinopathy among 13473 patients with diabetes mellitus in China: a cross-sectional epidemiological survey in six provinces. *BMJ Open* **7**, e013199 (2017). <https://doi.org:10.1136/bmjopen-2016-013199>
- 3 Yuan, B., Balabanova, D., Gao, J., Tang, S. & Guo, Y. Strengthening public health services to achieve universal health coverage in China. *Bmj* **365**, 12358 (2019). <https://doi.org:10.1136/bmj.12358>
- 4 Hou, X. H. *et al.* Data Resource Profile: A Protocol of China National Diabetic Chronic Complications Study. *Biomed Environ Sci* **35**, 633-640 (2022). <https://doi.org:10.3967/bes2022.078>
- 5 World Health Organization. *Consultation on development of standards for characterization of vision loss and visual function: WHO/PBL/03.91*, <https://apps.who.int/iris/bitstream/handle/10665/68601/WHO_PBL_03.91.pdf> (2003).

- 6 Cai, J. M. *et al.* Frequency of presenting visual acuity and visual impairment in Chinese college students. *Int J Ophthalmol* **13**, 1990-1997 (2020). <https://doi.org:10.18240/ijo.2020.12.22>
- 7 World Health Organization. *Global Physical Activity Questionnaire (GPAQ) Analysis Guide.*, <<https://www.who.int/docs/default-source/ncds/ncd-surveillance/gpaq-analysis-guide.pdf>>

Reviewer #2 (Remarks to the Author):

The results are noteworthy and will add significantly to the literature.

The results are relevant and compare well with the established literature.

The data support the conclusions. There are no significant flaws in the presentation of results. The methodology is sound and meets the relevant standards.

1. I would like to ask the question as to what percentage of patients had no gradable photographs and were they different from those who were included in the study? How would inclusion of those patients change the results?

Author response: Thanks for your comments. According to Supplementary Table 3, there are 1570 study participants (3.01%) with ungradable photos. Compared with those with gradable photos (n=50564, 96.99%), those with ungradable photos (n=1570, 3.01%) were older, having longer diabetes duration, and worse control of glycemia. Thus, the estimated DR proportion in this group might be a bit higher. However, due to the very low proportion of ungradable photos (3.01%) among all data, the effect was minimal. We have added the description of the comparison between those with ungradable photos and those with gradable photos in the result part, and a brief paragraph in the discussion part, as follows.

“Results” Line 110-114: “Compared with those with gradable photos (n=50564, 96.99%), those with ungradable photos (n=1570, 3.01%) were older, having longer diabetes duration, and worse control of glycemia (Supplementary Table 3). Thus, the estimated DR proportion in this group might be a bit higher. However, due to the very low proportion of ungradable photos (3.01%) among all data, the effect was minimal.”

“Discussion” Line 306-309: “Thirdly, the differences between those with ungradable photos and those with gradable photos might introduce selection bias. But due to the very low proportion of ungradable photos (3.01%), the effect was minimal.”

2. How were the graders trained and quality controlled? This has a bearing on the results as appropriate image analysis and quality assurance is the foundation of this work and so more details of image analysis is essential for this study.

Author response: Thanks for your comments. The team grading the photos in this study consisted of eight ophthalmologists, working in the ophthalmology center of the Shanghai Sixth People's Hospital affiliated to Shanghai Jiao Tong University School of Medicine. All of them received standard trainings before the survey, with a consistency rate of 85% achieved in a test of 50 standard DR photos. During the process of grading, each photo was graded independently by two ophthalmologists with masking methods. If there is inconsistency between the two ophthalmologists, a senior ophthalmologist will review the photos. Finally, at least two ophthalmologists reached an agreement for image results, and then their conclusions were adopted. We have added a part describing this in the method part as follows.

“Method” Line 372-378: “The team grading the photos in this study consisted of eight ophthalmologists, working in the ophthalmology center of the Shanghai Sixth People's Hospital affiliated to Shanghai Jiao Tong University School of Medicine. All of them received standardized trainings before the survey. Two qualified ophthalmologists graded each photograph, and a third ophthalmologist audited inconsistent results. Masking was adopted at each stage of evaluation.”

Reviewer comments, second round

Reviewer #1 (Remarks to the Author):

The comments have been satisfactorily addressed

Reviewer #2 (Remarks to the Author):

Thank you, my concerns are answered.